# Interventions for Preventing and Resolving Bullying in Nursing: A Scoping Review

**DOI:** 10.3390/healthcare12020280

**Published:** 2024-01-22

**Authors:** Corina Elena Luca, Alessia Sartorio, Loris Bonetti, Monica Bianchi

**Affiliations:** 1Department of Business Economics, Health and Social Care, University of Applied Sciences and Arts of Southern Switzerland, 6928 Manno, Switzerland; corinaelena.luca@eoc.ch (C.E.L.); alessia.sartorio@eoc.ch (A.S.); loris.bonetti@supsi.ch (L.B.); 2Regional Hospital of Lugano, Ente Ospedaliero Cantonale, 6900 Lugano, Switzerland

**Keywords:** bullying, incivility, nursing, prevention and control, workplace violence

## Abstract

Bullying in the workplace is a serious problem in nursing and has an impact on the well-being of teams, patients, and organisations. This study’s aim is to map possible interventions designed to prevent or resolve bullying in nursing. A scoping review of primary research published in English and Italian between 2011 and 2021 was undertaken from four databases (Cochrane Collaboration, PubMed, CINAHL Complete, and PsycInfo). The data were analysed using Arksey and O’Malley’s framework, and the Preferred Reporting Items for Systematic reviews and Meta-Analyses Extension for Scoping Reviews (PRISMA-ScR) Checklist was followed to report the study. Fourteen papers met the review eligibility criteria. The analysis revealed four main themes: educational interventions, cognitive rehearsal, team building, and nursing leaders’ experiences. Interventions enabled nurses to recognise bullying and address it with assertive communication. Further research is needed to demonstrate these interventions’ effectiveness and if they lead to a significant decrease in the short-/long-term frequency of these issues. This review increases the available knowledge and guides nurse leaders in choosing effective interventions. Eradicating this phenomenon from healthcare settings involves active engagement of nurses, regardless of their role, in addition to support from the nurse leaders, the organisations, and professional and health policies.

## 1. Introduction

Bullying, incivility and workplace violence are widespread issues in nursing [1]. Before approaching research projects or implementation studies on these phenomena, it is necessary to understand the meaning of the terms used in the literature to refer to them. Some definitions claim that bullying is persistent negative actions aimed at damaging the target’s professional and personal relationships through social exclusion and harassment [2], with unwanted, repeated, and harmful actions with the aim of humiliating, offending, and causing distress in the recipient [1]. Bullying can be carried out by managers or supervisors (vertical bullying) when managers do not recognise the abilities of employees, deprive them of career opportunities, and deny them promotion or training, or gossip to damage their reputation, or by colleagues (horizontal bullying) when a nurse is yelled at, belittled, or receives demeaning and impertinent remarks from colleagues, sometimes in front of other nurses, patients, and their families [3,4]. Lateral violence can occur as an isolated incident with no gradient of power between individuals (peers) in a shared culture. Conversely, bullying comprises repeated occurrences for at least six months [5]. Bullying and lateral violence share behaviours such as sabotage, internal fighting, scapegoating, and excessive criticism [2]. It has been hypothesised that workplace violence is rarely a sudden event, but rather the culmination of an escalation of negative interactions between people, beginning with low-intensity abuse typical of incivility [3]. Incivility is defined as “one or more rude, discourteous, or disrespectful actions that may or may not have negative intent” [1]. Bullying, lateral violence, and incivility are described as a continuum related to their frequency, intensity, and intent to harm the target [6], and are often used synonymously. Consequently, in this article, we will use the term bullying broadly, incorporating the meanings of incivility and lateral violence.

It should be considered that there are both personal and environmental and organisational factors that can facilitate incivility and bullying [7]. Among the personal factors there are a non-dominant ethnic background or non-native speaker in the country, nurses’ gender, being a young nurse, or having few years of experience in the current workplace [7,8]. Environmental and organisational factors are situation- or task-oriented leadership, lack of support, poor leadership, rigid hierarchical structures, informal organisational alliances, abuse of organisational procedures, increase in workload, understaffing, pressure placed on workers, and high levels of stress [7,8].

The prevalence of these phenomena in the nursing profession varies between 67.5% and 90.4% for incivility, while it is greater than 75% if among peers; lateral violence at 87%; and bullying at 81% [6]. This wide prevalence in nursing has led to bullying being considered an epidemic [9,10]; so, it is important that nursing leaders recognise the problem’s relevance, which, in addition to being widespread, has major impacts on co-workers, the work climate, patients, and organisations themselves [10]. Negative outcomes can occur in up to 75% of cases, with a wide range of psychological and physical symptoms and the feeling that work life exerts negative influences on private life [2,6]. These issues cause 29.9% of nurses to want to leave their jobs and 59.2% to leave the profession [2]; they can also affect patient care, compromising its quality and safety [2,11,12,13], and the organisational environment, leading to a poisoned work climate and large costs for absenteeism, illness, turnover, and decreased productivity [10,14].

Considering the impacts on the well-being of staff, patients, and organisations, nurse leaders must take responsibility by implementing interventions that enable its prevention or resolution. In this regard, to our knowledge, to date, there are not recent reviews of interventions and their effectiveness. With this scoping review, we aimed to find out more about possible preventive or resolving interventions for bullying in nursing.

## 2. Materials and Methods

### 2.1. Aim of Study

To map possible interventions designed to prevent or resolve bullying in nursing.

### 2.2. Research Design

A scoping review was conducted between April and July 2021, and followed the first five steps of the methodological framework proposed by Arksey and O’Malley [15]: (a) identify the review question on a broad domain of a discipline; (b) identify relevant studies; (c) study selection; (d) data charting; and (e) reporting results. The Preferred Reporting Items for Systematic Reviews and Meta-Analyses Extension for Scoping Reviews (PRISMA-ScR) Checklist was followed to report the study [16].

#### 2.2.1. Identifying the Review Questions

Are there interventions that enable the prevention of bullying in the nursing profession in healthcare settings?

Are there interventions that enable the resolution of bullying in the nursing profession in healthcare settings?

#### 2.2.2. Identifying Relevant Studies

We began by consulting a librarian for recommendations on the most relevant databases for this topic: the Cochrane Collaboration, PubMed, CINAHL Complete, and PsycInfo. MeSH and free terms were used, adapting them to the specific search methods of each database. The keywords bullying, lateral violence, horizontal violence, mobbing, workplace incivility, harassment, nursing, nurse, prevention, intervention, and solving were combined variously using the Boolean AND and OR operators, resulting in a search strategy that best answered the review questions (Table 1).

### 2.3. Study Selection

Two reviewers conducted the search simultaneously by applying predetermined inclusion and/or exclusion criteria to all papers independently at each stage of the selection process [17]. The inclusion criteria for articles were: (1) concerning bullying, lateral violence, or incivility between employees; (2) pertained to all healthcare settings and the nursing profession, specifically related to graduate nurses; (3) published between January 2011 and March 2021; (4) published in English and Italian; and (5) all study design types (quantitative, qualitative, and mixed-methods). Given the large number of empirical studies on the topic, we excluded conference papers, editorials, reports, books, and grey literature.

Each researcher conducted a selection process to determine article eligibility with an initial screening phase based on the information provided in the title and abstract, followed by mutual comparison and subsequent full-text screening, resulting in a classification of included, excluded, or uncertain studies. The comparison at the end of each stage aimed at maintaining an approach consistent with the review questions [17]. To resolve disagreements or doubts regarding the selected articles, the researchers consulted an experienced external researcher [17].

### 2.4. Data Charting

The authors constructed a tool that considered the elements of the review objective and question (author, year, country, objectives, study design and data collection instruments, participants/contexts, type of intervention, and key findings). This tool was used to graphically represent the data extraction process.

#### Ethical Considerations

As the scoping review did not involve human beings, the approval of an ethics committee was not necessary according to the Swiss Federal Human Research Legislation [18].

## 3. Results

Of the 1066 articles initially identified, after removing duplicates and studies considered irrelevant, 88 articles were selected for full-text screening and 14 met the review eligibility criteria. The selection process is depicted in Figure 1 [19].

The included studies (Table 2) were mainly conducted in the United States of America (twelve) and South Korea (two). The acute setting characterised all included studies, most of which were conducted in a single institution (ten), some in multiple institutions (two), and others were national surveys (two). Quantitative approaches were dominant (nine), followed by mixed-methods (three) and qualitative approaches (two). The main objectives common to the studies aimed at understanding the effectiveness of interventions by increasing the awareness and recognition of the phenomenon among nurses, reducing bullying in the analysed contexts, and acquiring knowledge and skills to deal with and respond to bullying situations. A qualitative study investigated the effectiveness of interventions based on nurse leaders’ experiences; another pursued the goal of understanding the prevalence of the phenomenon. The narrative synthesis of results identified four themes: educational interventions, cognitive rehearsal, team building, and nurse leaders’ experiences.

### 3.1. Educational Interventions

Some authors proposed educational interventions to address bullying [20,21,22,23,24]. These interventions considered the characteristics and consequences of bullying and were designed and carried out heterogeneously. Nikstaitis and Simko [23], starting with a literature review of the effects of incivility in the workplace and an overview of recommendations for a healthy work environment, stimulated a discussion among participants that included personal experiences, professionalism, attitudes, behaviours, and ways to prevent incivility. Howard and Embree [21] proposed an e-learning training, “Bullying in the Workplace: Solutions for Nursing Practice”, with content on bullying, reacting under stress, identifying conflict management styles, and creating safe environments. It was an online activity that used scenarios to enable participants to practise what they had learned. The use of scenarios for cognitive training of nurses to handle workplace bullying was also proposed by Kang and Jeong [22] in the form of a smartphone app, which included an introduction to nonviolent conversation as standard communication, six bullying scenarios, and a question and answer board. Chipps and McRury [20] followed up an educational moment on bullying with an online registry and checklist of negative behaviours for nurses to record behaviours observed or experienced during each shift over seven months.

Walrafen et al. [24] conducted a survey to determine the prevalence of horizontal violence that showed that the majority of participants witnessed/experienced eight of the nine behaviours associated with horizontal violence. They proposed a training program, “Sadly Caught Up in the Moment: An Exploration of Horizontal Violence”, which contained a review of each behaviour and appropriate responses.

### 3.2. Cognitive Rehearsal

Authors of four studies implemented cognitive rehearsal training, a communication technique taught to participants as a strategy to stop uncivil behaviour [25,26,27,28]. After a training intervention on incivility, Razzi and Bianchi [28] engaged participants in cognitive rehearsal training using cards with written responses to uncivil behaviour and providing examples of how to respond to such behaviour. This was followed by a role-play session in which they practised applying these responses. Kang et al. [26] investigated the effects of a cognitive rehearsal program on bullying among nurses using four phases. In the first phase, “scenario development”, nine bullying scenarios were created from the results of previous studies and interviews with nurses. In the “creation of communication standards” phase, participants made desirable communication for the scenarios by employing four components of the nonviolent communication technique. In the “role-playing” phase, they simulated the nine situations in a safe environment to express/manage the experienced anger, preventing the vicious cycle of bullying. Finally, in the “re-role-playing” phase, they developed cognitive training for means of coping transferable to similar situations in the future. Additionally, Kile et al. [27] proposed a training intervention on incivility with definitions, examples, ways of manifestation, and effects on nurses, patient safety, and organisations. They taught the cognitive rehearsal technique using visual cues written on cards to instruct participants on the main forms of incivility and appropriate responses. To personalise the training, they provided ten incivility scenarios specific to the care context in the role-play and application phases of cognitive rehearsal. Balevre et al. [25] started with a policy of non-tolerance of bullying and leadership empowerment as support for employee empowerment and structured training on the psychodynamics of bullying and coaching in cognitive rehearsal. Through cognitive rehearsal exercises and role-playing with scenarios designed to practise learned responses, they taught nurses defensive techniques against bullying. An effective and professional alternative to lateral violence for communicating needs, expectations, and conflicts was proposed by Ceravolo et al. [29] through the use of workshop moments to improve assertive communication skills, healthy conflict resolution, elimination of a culture of silence, and awareness of the impact of lateral violence.

### 3.3. Team Building

Some authors have proposed activities aimed at team building through member interactions [30,31,32]. Vessey and Williams [32], starting with a bullying situation, implemented a cognitive program in which each session included an overview of the objectives to be addressed, a brief review of the material, a didactic session, supportive experiential activities, and a group discussion. These sessions were held during morning meetings before patient care started through journal club activities.

Armstrong [30], through the Civility, Respect, Engagement in the Workforce (CREW) intervention, aimed to increase civility in the workplace as a response to what employee evaluations indicated about the interpersonal climate [34]. The four-week intervention included one meeting per week. In the first meeting, she used the “Anything Anytime” tool, which started with a discussion of a generic topic and enabled an understanding of group members’ varying perspectives. In the second meeting she used the “Geometry of Work Styles” tool, which requires participants to choose from four geometric shapes that relate to a personality type. On the third day, using cues from nursing research, she stimulated a discussion on the definition and characteristics of incivility and assertive responses to it. Finally, participants practised actively replying to incivility scenarios in an interactive and safe setting. Each session concluded with a discussion of how a civil workplace can be reached, regardless of individual differences. Keller et al. [31] explored the perceptions, attitudes, and experiences of nurses who completed the Bullying Elimination Nursing in a Care Environment (BE NICE) Champion program. This program taught them how to recognise signs of bullying and provide support to their peers, facilitating the creation of bullying intervention strategies through didactic training and role-plays simulating bullying scenarios and the correct way to deal with them using the 4S strategy. The first S, “Stand by”, requires facilitators to be close to the bullying victim to convey the message that they are not alone. “Support” implies that facilitators show empathy, actively listen, and acknowledge the victim’s feelings. Involved people who report bullying to nurse leaders apply the “Speak up” component of the 4S strategy. “Sequester” implies that facilitators remove the victim from the situation.

### 3.4. Nursing Leaders’ Experiences

Skarbek et al. [33] highlighted which interventions are considered effective in addressing bullying from nurse leaders’ perspectives. While institutional “mandatory programs” are not perceived as effective, nurse leader-initiated individual unit-level interventions, in collaboration with administrative and institutional support, were seen as effective ways to address bullying. They agree that to establish a healthy work environment, the behavioural characteristics of collaboration, respect, effective interpersonal communication, collegiality, and mutual support must be evident to those entering the profession, senior nursing staff, and nurse leaders to build positive social practices.

## 4. Discussion

The magnitude of bullying in nursing has led nurse leaders to question more about the extent of the phenomenon within their own institutions, an aspect confirmed by an exponential increase in publications in recent years. Bullying often occurs with a peer form of hostility towards novices, but nurses with more years of service and nurse leaders are also exposed to this phenomenon [2,9,35,36]. Therefore, it is necessary to know if there are interventions to prevent or resolve bullying among nurses in healthcare settings. The findings from the 14 identified studies highlight different interventions designed with the aim of testing their effectiveness in addressing and curbing bullying. Despite the heterogeneity of the proposed interventions, the common goal was to increase the awareness and recognition of bullying among nurses, develop the ability to respond assertively to uncivil behaviour, and reduce bullying in the analysed contexts. Educational interventions have been offered in the form of training sessions [20,23,24], e-learning [21], and a smartphone application [22]. Some of these facilitated knowledge creation about bullying through case discussions, literature reviews, and discussions of uncivil behaviour and consequences [20,23]. Others have found it necessary to increase knowledge regarding types of communication, such as conflict management, crucial conversations, and nonviolent communication [21,22], and still others have used prevalence results on lateral violence to create training on the behaviours that emerged from the study [24]. In terms of evaluating intervention effectiveness, post-intervention measurements have been used that have given varying results, including an increase in perceptions and experiences of bullying; this is considered a positive indicator as it allows for the identification of negative behaviours and increased awareness [20,23]. Evaluations of the educational interventions revealed their influence on communication skills, which resulted in a positive effect on conflict management strategies among nurses and decreases in work-related bullying experiences and turnover intention [21,22]. The impact of the educational program on the behaviours noted in their own care settings [24] has been linked to the development of dialogue among nurses and their sense of professional responsibility, which are useful in breaking the cycle of horizontal violence in work environments.

Most of the included articles described cognitive rehearsal and team-building interventions aimed at fostering communication and interaction among employees at work [25,26,27,28,29,30,31,32]. Cognitive rehearsal [25,26,27,28], the most widely used intervention, is a therapeutic technique in which an individual imagines situations that tend to produce anxiety or self-destructive behaviours and then repeats positive coping statements or mentally rehearses a more appropriate behaviour [37]. For its implementation, the authors used bullying scenarios, provided positive coping responses to those scenarios, and included role-play in which participants could practise the learned responses [25,26,27,28]. An evaluation of its effectiveness has shown that this approach improves interpersonal relationships, trains people to cope with bullying, decreases turnover intention [26], causes a perceived change in group behaviour in dealing with bullying, creates positive cultural change [25], and results in an increase in the ability to both recognise incivility and deal with it [27], an increase in awareness of incivility [28], and a reduction in the incidence of exposure to incivility [27,28]. Another type of intervention is related to healthy conflict resolution through assertive communication and eliminating the culture of silence among nurses [29].To achieve this, nurse leader-focused workshops were held in which their roles in demonstrating learned behaviours to employees was emphasised, followed by interventions to foster peer learning. The effectiveness of this intervention was observed in decreased verbal abuse, increased perception of a respectful workplace, and a higher rate of nurses determined to solve the issue after an episode of lateral violence.

Finally, team-building interventions have been proposed in different formats and settings. Vessey and Williams [32] presented a cognitive program starting from an actual bullying case, integrating discussions on the topic and experiential and journal club activities into daily nursing unit meetings. Armstrong [30] adopted the CREW method with the goal of team building and creating awareness of how a civil workplace can be achieved, regardless of individual differences. Keller et al. [31] emphasised the recognition of bullying and peer support using the 4S intervention to convey messages of closeness to the bullying victim, to show active listening, encouraging the reporting of bullying to superiors, and actively intervening when it occurs to remove the victim from the situation, discouraging the vicious cycle of the phenomenon. The effectiveness of team-building interventions has been demonstrated through the detection of positive and proactive engagement among participants [32], an increase in nurses’ competence to recognise workplace bullying, and the ability to respond when it occurs [30,31].

In addition, this review found that nurse leaders’ organisational engagement and support, through behaviours that model for co-workers, is a vital component of empowerment and is crucial and effective in addressing workplace bullying [25,29,31,33], and that it is important to intervene at all levels (society/policy, organisation/employer, work/task, and individual/work interface) to prevent it [38]. In contrast, implementing zero-tolerance policies and passive dissemination of information about bullying have proven ineffective [39].

The studies predominantly considered interventions implemented in the acute hospital setting to address a problem present in the analysed settings evidenced by pre-intervention measurements [20,21,22,23,24,26,27,28,29]. Although the implementation of these interventions was related to solving the problem, they are believed to have the potential to reduce bullying and consolidate a positive and civil culture in the workplace, and can be used with preventive intent and implemented by all nurses (novice and experienced).

## 5. Strengths and Limitations

Among the main strengths of this review are the adoption of a reproducible method and the systematic approach. To reduce the possibility of selection bias to a minimum, the study selection procedures described in the methods were rigorously adhered to. In accordance with the methodology used, a quality appraisal process was not performed on the included studies.

The geographic concentration of studies in only two countries may limit the transferability of results to other health systems, as bullying tends to be related to the culture of the setting. Limiting the review to articles published in English and Italian and to literature published in databases may have led to an incomplete overview of available data and knowledge that could have added information to this review. However, the researchers chose to include articles describing research projects on the topic to identify effective interventions to counter bullying, with a view to future research projects.

## 6. Conclusions

This review revealed that several interventions have been designed to address the problem of bullying among nurses in healthcare settings by implementing educational, cognitive, and empowering interaction approaches among team members. Although the results showed the effectiveness of the interventions concerning nurses’ recognition of the phenomenon and increased skills in addressing it with assertive communication, only a slight and not always significant reduction in the presence of this phenomenon was observed. Consequently, new research projects are necessary to demonstrate the effectiveness of the interventions, including healthcare settings other than those where they have been implemented so far, robust study designs, such as RCTs, to assess their real effectiveness and adaptability to the context, and to understand whether the effects persist over time, leading to a significant decrease in the frequency of bullying among nurses. It will, moreover, be the responsibility of each nurse leader to identify the intervention that best fits their context.

Nurse leaders play a crucial role in preventing bullying in care settings. They should have a thorough understanding of the manifestations of the phenomenon and its consequences to recognise timely dysfunctional relational dynamics arising in their own teams and with peer leaders or superiors. Nurse leaders also have a responsibility to recognise and address personal, environmental, organisational, and cultural factors that may facilitate bullying in their context. This review helps to increase available knowledge on the topic and guides nurse leaders in choosing effective interventions to be adapted and implemented in the specific context. It also raises their awareness of the importance of leading by example, recognising their teams’ relational patterns, and discouraging hostile peer interactions as preventive actions to foster cultural change in their context. Eradicating this phenomenon in healthcare settings involves active engagement of nurses, regardless of their role, in addition to support from the nurse leaders, the organisations, and professional and health policies.

## Figures and Tables

**Figure 1 healthcare-12-00280-f001:**
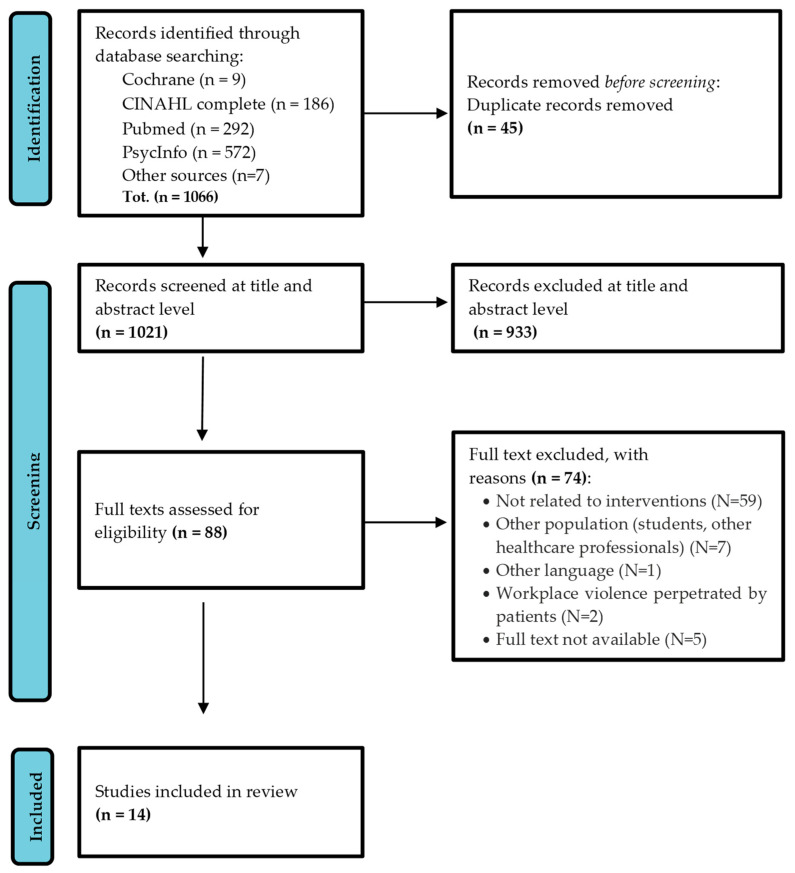
PRISMA statement.

**Table 1 healthcare-12-00280-t001:** Search strategy.

Databases	Search Strategy	Boleean Operators
Cochrane Collaboration Pubmed CINAHL Complete PsycInfo	bullying lateral violence horizontal violence mobbing workplace incivility harrasment	OR
	nurs *	AND
	intervention	AND

(*) Truncation symbol.

**Table 2 healthcare-12-00280-t002:** Study characteristics.

Intervention Category	Author(s)/ Year/Country	Aims	Study Design and Data Collection Methods	Participants/ Settings	Intervention (Method, Contents, Intensity/Duration)	Key Findings and Outcomes
**EDUCATIONAL INTERVENTIONS**	**Chipps and Mc Rury 2012 [20]** United States	The purpose of this pilot study was to describe the effect of an educational program on workplace bullying provided to nursing staff and examine its’ impact on reducing workplace bullying on nursing units.	The design of this pilot study was a quasi-experimental pretest and posttest comparison using Negative Acts Questionnaire-Revised (NAQ-R) and questions related to job satisfaction, errors, and near errors in clinical practice and patient satisfaction.	Sixteen staff members on inpatient hospital nursing units.	The content of the educational programme on bullying in the workplace focused on identifying bullying behaviour and understanding the consequences of bullying behaviour. After the programme, a sample of staff agreed to participate in the pilot research study. Staff were given a Web-based logbook that contained a checklist of negative behaviours that were observed or experienced during each shift for the 7-month study period.	Thirty-seven percent of respondents self-identified as having experienced bullying at least weekly. This decreased to 6% post-intervention. Respondents were asked, on the prettest and the posttest, if they had been witnessing to any bullying in the workplace. 75% of respondents on the prettest indicated that they had witnessed bullying. This increased but not significantly to 88% on the posttest. This finding may suggest that the intervention was effective in assisting individuals to identify negative behaviors.
	**Howard and Embree 2020 [21]** United States	The purpose of this study was to examine whether an educational intervention can increase awareness and knowledge of incivility and bullying and enhance communication skills.	This study used a pretest-posttest quasi-experimental mixed- method design. The Workplace Civility Index (WCI) was administered to measure the effectiveness of the intervention.	Participants included 49 nurses from an academic medical center in the Midwestern United States.	The educational intervention used, entitled ‘Bullying in the Workplace: Solutions for Nursing Practice’, was offered in the form of e-learning developed in conjunction with Sigma Theta Tau International Honor Society of Nursing (Sigma, St. Louis, MO, USA). The online educational activity required approximately 2.5 h and focused on defining the phenomenon, reacting under stress, identifying conflict management styles, creating a safe environment and how to hold crucial conversations. Because the intervention was an online learning activity, the communication activity used scenarios to provide participants with the ability to practice what they learned.	Within the experimental group, all participants noted the successful use of a positive conflict management strategy after the educational intervention. This study provided evidence to support the efficacy of an asynchronous provider-directed, learner-paced e-learning educational activity in decreasing incivility and increasing perceived comfort level during critical conversations between nurses.
	**Kang and Jeong 2019 [22]** South Korea	To develop a cognitive rehearsal intervention for workplace bullying and examine its effects on nurses’ bullying experiences and turnover intentions.	Cluster quasi-randomized trial. Negative Acts Questionnaire-Revised (NAQ-R) validated for Korean nurses with a modified version of “intent to quit”.	Participants included 72 hospital nurses working in a university hospital in South Korea.	The intervention was offered in the form of a smartphone application to cognitively train nurses to deal with bullying situations in the workplace. The application consists of an introduction to non-violent conversation as standard communication, six scenarios of bullying situations and a question and answer board. The intervention group was introduced to non-violent communication and training in the application usage for 2 h, at which time the researchers installed the application on the smartphones of the intervention group. During the 8-week intervention period, push alarms were sent twice a day to encourage the use of the application.	The cognitive rehearsal intervention developed in this study was effective for decreasing nurses’ person-related bullying, work-related bullying experiences, and turnover intention. However, it had no effects on intimidation related bullying experiences.
	**Nikstaitis and Simko 2014 [23]** United States	To determine if a nursing education program would increase awareness of incivility and impact the number of perceived incidences by (1) assessing nurses’ experience of incivility along with discerning the perceived source of the incivility and (2) educating the nurses, thus determining if the in-service education decreases the incidence of incivility.	This is a quantitative pilot study that utilized a 1-group preintervention and postintervention test design. Nursing Incivility Scale was used as measurement tool.	The inclusion criteria were the Saint Agnes adult in ICU. A total of 21 nurses completed all parts of this study.	The education sessions for the nurses included case studies, review of the literature about the effects of incivility in the workplace, and an overview of recommendations for a healthy work environment along with resources for the nurses. Then a facilitated discussion was conducted describing personal experiences of nurses in the adult ICU setting. This included discussions about professionalism, behaviors, attitudes, and ways to prevent workplace incivility. Five education sessions of one hour took place.	The postintervention score had a higher mean than the preintervention score in each of the dimensions. Higher scores indicate incivility; thus, lower scores indicate civility. Therefore, more instances of incivility were identified after intervention to increase awareness of incivility. The results of the current study found that incivility perceptions were higher in the postintervention survey; these findings suggest that the education was effective, thus creating more awareness of incivility.
	**Walrafen et al. 2012 [24]** United States	The purpose of the study was to determine the prevalence of horizontal violence in a multi-institutional hospital system.	A mixed-method descriptive design was used, using the Horizontal Violence Behaviour Survey and the participants were asked to respond to three open-ended qualitative questions.	All nurses in the multi-institutional health care system were invited to participate in the study. The respondents were 227 nurses.	A 30-min educational program entitled “Sadly Caught Up in the Moment: An Exploration of Horizontal Violence” was developed that focused on heightening awareness by providing examples of negative behaviors. The intervention was composed of a review of each of the behaviors including appropriate responses when the behaviors were encountered. Additional components of this program included a review of available resources within the organization as well as the role of resilience in helping individuals deal with adversity. Since the development and offering of the educational program, 700 nurses in the organization have attended.	While the major aim of this study was to determine the prevalence of horizontal violence within the organization, the findings clearly called for the development of an intervention to address this phenomenon. As a result of the educational program, a dialogue has begun among the nurses within the organization, focused on encouraging an increased sense of professional accountability among nurses to break the cycle of horizontal violence in their individual work environments.
**COGNITIVE REHEARSAL**	**Balevre et al. 2018 [25]** United States	This article describes a nursing professional development evidence-based intervention project addressing the significant problem of bullying in the nursing workplace	The measurement design for the project was a nonexperimental, descriptive, pre and post intervention comparison, using the modified Organizational Tolerance for Sexual Harassment Inventory renamed the WHS-2013.	A 30-bed, medical-surgical nursing unit was selected for the project and had over 50 nursing staff employees.	The project lasted 9 weeks and included four interventions: (a) policy enforcement training, (b) psychodynamic education about bullying, (c) cognitive rehearsal training and coaching, and (d) empowering leadership support. After reviewing the hospital’s no-tolerance policy on bullying, the project’s instructional design guided NPD education about the psychodynamics of bullying. Training exercises in cognitive rehearsal were implemented to teach the nursing staff techniques to defend against bullying, and scenarios were incorporated through which participants could practice their learned responses, making them scripted reactions. These exercises were designed to change thinking and behavior, thereby creating empowerment, self-efficacy, and collegiality. The leadership support was a vital component of empowerment. The bulk of the project’s education centered on cognitive rehearsal training and role-playing.	Results demonstrated statistically significant t-test comparisons of pre- and postsurvey measures, supporting the clinical questions that empowerment and perceptual change drove individual and group behavior to confront bullying and create a positive culture shift. This phenomenon was especially evident when the staff reported events in which they had successfully used scripting from the cognitive rehearsal training both as individuals and in teams. The unit’s culture began to change from one of incivility to one that was resilient to adversity through cognitive reframing.
	**Kang et al. 2017 [26]** South Korea	This research aimed to investigate the effects of a cognitive rehearsal program (CRP) on workplace bullying among nurses.	A randomized controlled trial was performed. Interpersonal relationships (Relationship Change Scale), workplace bullying (Negative Acts Questionnaire-Revised), symptom experience (Brief Symptom Inventory), and turnover intention (nurse turnover intention tool) were measured at pre and post-intervention.	Participants were 40 nurses working in different university hospitals in B city, South Korea.	The intervention was developed using the 4 stages of Cognitive Rehearsal Programme (CRP) and the nonviolent communication technique. In the first stage, “developing scenarios”, were created 9 types of workplace bullying situations based on previous studies and nurse interviews. “Creating communication standards” involved designating what constitutes desirable communication for the scenarios, employing 4 components of nonviolent communication technique: “observation”, “feeling”, “need”, and “request”. In “role-playing”, subjects act out the 9 situations in a safe environment, which helps them express and handle the anger and suppression that the subject experienced before. After revising “communication standards” based on the nonviolent communication techniques they learned, “re-role-playing” was performed to practice them. “Feedback and evaluation” of the final stage, was performed in every session. In this study CRP comprised 10 sessions in a total of 20 h for 5 weeks. Each session took 2 h.	The CRP for workplace bullying improves interpersonal relationships and decreases turnover intention. So, it can be utilized as one of the personal coping strategies to reduce the turnover among nurses.
	**Kile et at. 2019 [27]** United States	The aim of this study was to assess the effect of education and the use of cognitive behavioral techniques on registered nurses’ ability to recognize and confront incivility and its effect on job satisfaction.	Pilot study, mixed method. Nurse Incivility Scale (NIS), the NDNQI Index of Work Satisfaction Nurse Interaction subscale, and two open-ended questions.	Participants recruited are 17 nurses from the Post-Anesthesia Care Unit (PACU) in a 238-bed, ANCC Magnet^®^ designated rural community hospital located in Virginia.	The program was composed of five training sessions approximately 2 h in length that took place over three weeks. The first hour of training included a didactic session providing definitions and examples of incivility and the different ways it can manifest and potential effects of incivility on nurses, patient safety, and organizations. Participants were instructed on the top 10 forms of incivility and the appropriate cognitive rehearsal techniques for responding to each of these. This education included the use of cue cards containing written visual cues for the appropriate responses to the most common uncivil behaviors. The second hour of the sessions involved role-playing. People learn from one another via observation, imitation, and modeling. Scenarios which exemplified each of the top 10 forms of incivility were specific to a PACU. These scenarios included the appropriate cognitive behavioral technique to be used when addressing each type of uncivil behavior.	Two subscale means were significant. The remaining NIS subscale means and the NDNQI Nurse Interaction subscale decreased across three time points (initial, immediate postintervention and final survey conducted six weeks postintervention). Qualitative data supported findings. NDNQI Index of Work Satisfaction had no effect on nurse job satisfaction. The intervention was effective in increasing nurses’ recognition of incivility and ability to confront it. Perceived instances of incivility decreased over time.
	**Razzi and Bianchi 2019 [28]** United States	The purpose of this quality improvement program was to develop an educational module designed to increase the awareness of incivility in the workplace and train the participants to respond to incivility using cognitive rehearsal.	A quality improvement program was conducted and Nursing Incivility Scale was used as measurement tool.	The sample size included 24 participants from a 232-bed community hospital located in the Northeastern United States was used.	A 1-h educational program was conducted in several sessions. The background and significance of incivility was discussed to increase the participants’ understanding of the magnitude of incivility in nursing. The discussion also included content about interventions, other than cognitive rehearsal. Cognitive rehearsal training was provided at the end of the education session. First, participants were educated on what cognitive rehearsal is, why it is effective, and how it is used to combat incivility. Next, the participants were given cue cards with scripted responses to uncivil behavior; they were given some examples of how they could respond to certain behavior using these scripted responses. The participants then had time for role playing to practice using the scripted responses.	Findings highlighted that participants had an increased awareness and a decreased incidence in exposure to incivility, theoretically due to their responding to incivility with more effective communication. Incivility programs can provide nurses with the needed tools to identify uncivil behaviors and react in a proactive, professional manner; this promotes a safe working environment for nurses and their patients.
	**Ceravolo et al. 2012 [29]** United States	This quality improvement project aims to reduce nurse-to-nurse lateral violence and create a more respectful workplace culture, to enhance assertive communication skills and raise awareness through a series of workshops.	This quality improvement project describes the organization-wide pre- and post-intervention survey of registered nurses’ perception of lateral violence and turnover. Survey items were adapted from the Verbal Abuse Survey.	Five-hospital integrated health-care delivery system in the north-eastern USA were involved from 2008 to 2011 (203 workshops and over 4000 nurses).	The 60- to 90-min workshops were designed to enhance assertive communication skills and raise awareness about the impact of lateral violence behaviour. Emphasis in all of the workshops was placed on healthy conflict resolution and eliminating a culture of silence for nurses. Helpful acronyms as memory aides were shared and practiced to strengthen effective communication and conflict resolution. The memory aides and acronyms were designed to assist nurses to standardize communication about their concerns and needs in a succinct and assertive manner. The tools provided a professional and effective alternative to using lateral violence to communicate expectations, needs and conflicts	After the workshop series, nurses who reported experiencing verbal abuse fell from 90 to 76%. A greater percentage of nurses perceived a workplace that was respectful to others and in which it was safe to express opinions. After the workshop series, a greater percentage of nurses felt determined to solve the problem after an incident of lateral violence, while a smaller percentage felt powerless. Nursing turnover and vacancy rates dropped.
**TEAM BUILDING**	**Armstrong 2017 [30]** United States	The purpose of this quality improvement project was to see if implementation of a civility training program would: (a) increase the staff nurses’ ability to recognize workplace incivility (b) reduce workplace incivility on a nursing unit (c) increase confidence in the staff nurse ability to respond to workplace incivility when it occurs.	Pilot study using CREW intervention with pretest and posttest questionnaires (Workplace Incivility Scale and the Confidence Scale).	The project was implemented in a medically focused medical surgical unit at a rural Kentucky hospital with a sample of nine registered nurses.	The intervention used was Civility, Respect, Engagement in the Workforce (CREW), which lasted four weeks with one meeting per week. The session on the first day included the Anything Anytime tool, which consists of providing a generic topic and discussing how it is viewed differently by different group members. The second day’s session involved the Geometry of Work Styles tool, which requires participants to choose from four geometric shapes that relate to a personality type. Day three included a facilitated discussion on the definition, characteristics and how to respond assertively to incivility effectively using insights from nursing research. Participants practiced actively responding to incivility scenarios provided by the facilitator in a safe but interactive environment. Each facilitated discussion concluded with a discussion of how a civilised workplace can be achieved despite individual differences.	None of the posttest items of the Workplace Incivility Scale had statistically significant differences when compared with the pretest. With regard to the Confidence scale the analysis revealed a statistically significant increase in the posttest mean scores for each item on the instrument, when compared to the mean scores on the pretest. Result in statistically significant increases in the nurses’ self-assessed ability to recognize workplace incivility and confidence in the nurses’ ability to respond to workplace incivility when it occurs.
	**Keller et al. 2019 [31]** United States	To explore the experiences, perceptions and attitudes of registered nurses (RNs) who completed the BE NICE Champion training programme.	A qualitative focus group approach.	The participants were 25 nurses from the Departments of Nursing at Tisch Hospital, NYU Langone Health’s flagship acute-care facility.	The proposed programme is Bullying Elimination Nursing in a Care Environment (BE NICE) lasting 1.5 to 2 h, in which the skills needed to identify the signs of bullying and how to support their peers are taught, leading to the creation of the 4S bullying intervention strategies. The first strategy, “Stand by,” requires champions to physically stand near the bullying victim, conveying the message that they are not alone. “Support” implies that champions actively listen, show empathy and acknowledge the victim’s feelings. Those involved reporting bullying to their nurse leaders, with or without the victim present, demonstrate the “Speak up” component of the 4Ss. Finally, “Sequester” involves champions removing the victim from the situation, creating a safe and supportive environment for the victim, while discouraging future acts of aggression for the bully.	Three consistent themes emerged from a content analysis of the transcripts: (a) awareness and understanding, (b) applying the 4S’s and (c) feeling prepared and empowered. Additional themes included impact on the work environment and additional programme recommendations tailored to nursing leadership. The programme and use of the 4S’s adequately provided RNs with confidence to intervene when bullying was observed. RNs felt better equipped to respond due to the techniques learned and appreciated the organizational commitment to address bullying.
	**Vessey and Williams 2021 [32]** United States	The overall goals of the program were 3 pronged: staff education recognizing behavioral and cultural “red flags,” and empowering staff to do the right thing by actively embracing changes that strengthen the unit’s climate and improve teamwork.	Quality improvement project The Plan, Do, Study, Act (PDSA) framework served as the template for this QI project. Negative Acts Questionnaire-Revised (NAQ-R) was used as measurement tool.	All nurses and ancillary department staff (N = 16) working in the clinical research unit participated.	The final project consisted of 6 sessions with each session rolled out at 2- to 3-week intervals. Each session consisted of (1) a 30- to 60 min in-service group activity embedded into standing unit meetings, (2) online journal club readings, and (3) morning huddles prior to when care of study patients commenced where the key information from the group activities and readings was discussed and reinforced. Didactic content on BLV was culled from the empiric and policy literature and additional pedagogical resources were identified. Each session began with an overview of the objectives to be covered, a brief review of previous material, and a short didactic session and supportive experiential activities, followed by group discussion.	Throughout this project, it was clear that the topic of BLV was meaningful to participants. For some of the experienced nurses, the information helped explain and label incidents they may have encountered earlier in their careers. For professionally newer nurses, it helped them realize that the problem of BLV had significant roots in health care and nursing culture, providing a framework for negative encounters they may experience or witness. Collectively exploring BLV, individual response patterns, peer support, and hospital and professional resources contributed to greater positive engagement and proactive interventions with each other.
**NURSING LEADERS’ EXPERIENCES**	**Skarbek et al. 2015 [33]** United States	The aim of this study was to acquire nurse managers’ perspectives as to the scope of work-place bullying, which interventions were deemed as effective and ineffective, and what environmental characteristics cultivated a healthy, caring work environment	Qualitative design In-depth, semistructured interviews. Ray’s theory of bureaucratic caring guided the study.	The participants were six nurse managers who were employed in 4 urban hospitals and 1 suburban hospital in the Midwestern and Northeastern regions of the United States.	No intervention has been performed.	NM-initiated interventions on a unit level, in collaboration with institutional and administrative support, were perceived to be effective methods to address workplace bullying among RNs. Communication, collective support, and teamwork are essential to create environments that lead to the delivery of safe, optimum patient care.

## Data Availability

No new data were created or analyzed in this study. Data sharing is not applicable to this article.

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
