# Peer review of "Interventions for Preventing and Resolving Bullying in Nursing: A Scoping Review"

_healthcare, 2024, doi:10.3390/healthcare12020280_

Round 1
Reviewer 1 Report
Comments and Suggestions for Authors
A very precise and readable paper. The introduction and the methodology section have been clearly explained. A few questions, if considered appropriate will lend further value to the paper.
1.Rationale behind the time frame chosen for Scoping Review?
2.The words used in th ereview as synonyms have been established as separate constructs already in scholarly literature. What , then, is the rationale for using them as synonyms?
3.The research question has mixed content. The nature of 'preventive ' and ,resolving' interventions would be different. Former would follow a proactive approach and the latter a reactive one. Shouldn't two separate research questions be framed?
4.The statistics provided in the introduction section pertain majorly to 2017-2019. The authors are making a statement that the prevalence of workplace bullying research in increasing exponentially(line 252)Latest data in that case should have been incorporated in the introduction i. e 2022,2023.
5. PRISMA statement as Fig 1 is not visible,neither in the main text or as supplementary(page 3,line116)
Comments on the Quality of English LanguageThe quality of english is fine and readable. Minor errors in sentence construction may be corrected.
Author Response
Dear Editor,
Thank you for giving us the opportunity to revise the manuscript.
Please find in the file upload below our point-to-point responses to the comments made by the reviewer 1.
Kind regards,
The Corresponding Author

Reviewer 2 Report
Comments and Suggestions for Authors
This is a review of previous published studies.
Intro: Authors reviewed definitions of bullying/incivility behaviours - which is important as these often overlap or are not clear to readers.
Methods: clearly stated. Table 2 helpful particularly if reader wants to look at a specific study included in the review. Need more information regarding whether the interventions change negative behaviours. Could not find Fig. 1
Results: Although the authors state that most of the studies were quantitative - this is not really accurate. There were no direct measurements of number of bullying incidents (such as number of reported incidents) but rather results relied on nurses remembering whether there had been a decrease in incidents - relying on nurses' perceptions. Educational interventions focused on recognition of incidents and improving communication styles. Cognitive rehearsals were to help improve response communications to de-escalate a situation. Team building focused on support of the victim at the time of the incident and reporting the incident. Nursing leaders experiences looked at initiating interventions to create a healthy work environment.
DISC: The authors report a modest benefit in decreasing bullying etc incidents when using these various techniques.
CONCLUSION: The authors conclude "new research projects are necessary to demonstrate the effectiveness of the interventions...."and it is the "responsibility of each nurse leader, to identify the intervention that best fits their context."
CRITIQUE: The authors reviewed available studies looking at interventions to decrease incivility and bullying behaviours. However, the onus of decreasing these incidents is largely placed on the victims and their co-workers - ways to respond to de-escalate a situation. Only the nursing leaders experiences address the workplace environment.
The techniques reviewed in this study have value - techniques to de-escalate a situation are important, therefore, publishing this review will be valuable to health workers dealing with these situations.
However, what really needs to be done is to identify the institutional, organizational and cultural stressors that lead to bullying (too many patients per health worker, long work hours, racism and classism in the institution and the culture, inadequate financial and emotional support, etc) that causes these negative behaviours. I think the authors should, at least, mention these and the need to investigate and ameliorate them.
Author Response
Dear Editor,
Thank you for giving us the opportunity to revise the manuscript.
Please find in the file upload our point-to-point responses to the comments made by the reviewer 2.
Kind regards,
The Corresponding Author

Reviewer 3 Report
Comments and Suggestions for Authors
Dear Authors,
I am delighted with your manuscript.
It is perfect, exemplary. It is an example of a job very well done.
Starting from the selection of the topic, through conducting the research and drawing conclusions, and selecting the literature - I rate everything as the highest.
Maybe I would change the topic a bit to make it more focused on the workplace and not nursing. The current wording of the title may also incorrectly suggest patient bulling.
GOOD JOB
Author Response
Dear Editor,
Thank you for giving us the opportunity to revise the manuscript.
Please find in the file upload our point-to-point responses to the comments made by the reviewers 3.
Kind regards,
The Corresponding Author
